# Why is reporting quality improvement so hard? A qualitative study in perioperative care

Emma Leanne Jones,[1,2] Mary Dixon-Woods,[3] Graham P Martin[3,4]

¹Clinical Trials Unit, University of Warwick, Coventry, UK
²Orthopaedic Directorate, University Hospitals Coventry and Warwickshire NHS Trust, Coventry, UK
³THIS Institute (The Healthcare Improvement Studies Institute), University of Cambridge Primary Care Unit, Cambridge, UK
⁴Health Sciences, University of Leicester, Leicester, UK

**Correspondence to**
Dr Emma Leanne Jones;
e.jones.11@warwick.ac.uk,
embosymes@yahoo.co.uk

## ABSTRACT

**Objectives** Quality improvement (QI) may help to avert or mitigate the risks of suboptimal care, but it is often poorly reported in the healthcare literature. We aimed to identify the influences on reporting QI in the area of perioperative care, with a view to informing improvements in reporting QI across healthcare.

**Design** Qualitative interview study.

**Setting** Healthcare and academic organisations in Australia, Europe and North America.

**Participants** Stakeholders involved in or influencing the publication, writing or consumption of reports of QI studies in perioperative care.

**Results** Forty-two participants from six countries took part in the study. Participants included 15 authors (those who write QI reports), 12 consumers of QI reports (practitioners who apply QI research in practice), 11 journal editors and 4 authors of reporting guidelines. Participants identified three principal challenges in achieving high-quality QI reporting. First, the broad scope of QI reporting—ranging from small local projects to multisite research across different disciplines—causes uncertainty about where QI work should be published. Second, context is fundamental to the success of a QI intervention but is difficult to report in ways that support replication and development. Third, reporting is adversely affected by both proximal influences (such as lack of time to write up QI) and more distal, structural influences (such as norms about the format and content of biomedical research reporting), leading to incomplete reporting of QI findings.

**Conclusions** Divergent terminology and understandings of QI, along with existing reporting norms and the challenges of capturing context adequately yet succinctly, make for challenges in reporting QI. We offer suggestions for improvement.

## INTRODUCTION

Quality improvement (QI) seeks to improve the functioning of healthcare organisations by making systematic improvements to healthcare systems and processes.[1 2] We previously, in the context of a systematic review,[3] defined QI as involving both QI methods, including approaches such as Plan-Do-Study-Act cycles, Lean and Six Sigma,[1 4] and QI interventions, which are specific activities, actions or instruments targeting defined areas of practice (eg,

### Strengths and limitations of this study

► This is among the first studies to examine the influences on reporting of quality improvement in healthcare.
► An international and multidisciplinary study, it offers specific insights in the area of perioperative care.
► Participants offered suggestions for improving reporting of quality improvement (QI) in perioperative care specifically, which may have relevance for other clinical fields.
► This study does not include patients as a QI stakeholder group.
► This study recruited only stakeholders who were actively interested in QI.

checklists).[5] Despite the growing use and popularity of QI and its potential to benefit patient care, the academic literature on QI is itself problematic. One major problem relates to quality of reporting of QI in the academic literarure.[6] This is a problem that, as in other fields, limits the inferences that can be drawn, impairs confidence in the findings and thwarts the ability to replicate and scale.[7 8]

Some possible reasons for the poor quality of reporting of QI in the academic literature likely relate to the distinctive nature of the interventions and methods used in QI, which often evade straightforward description, not least because of their adaptive and iterative character.[9–12] Similarly, the mechanisms through which interventions work, often sociotechnical in nature, may not be easily visible and may be difficult to account for.[13 14] The variability of QI adds to the complexities: QI may be conducted in a variety of forms, from improvement projects led by local clinicians in a single setting through to multisite research using experimental designs.[15–17]

Efforts to improve reporting of QI include the Standards for QUality Improvement Reporting Excellence (SQUIRE) 2.0[10] guidelines, but problems nonetheless remain. A systematic review of QI reporting

**BMJ**

in perioperative care, for example, showed that 74% of publications fail to adequately describe implementation fidelity, 73% do not describe how interventions were modified, and 62% omit details of the materials needed to replicate the intervention.[3] Moreover, reporting guidelines are not a panacea: they can codify what should be included, but may be less useful in influencing how well these things are reported. For example, checklists may prioritise mechanistic compliance over rich and detailed reportage[18] and authors may not have adequate training on how to use reporting guidelines[19] and may not know which one is most appropriate for their study.[20] The question of what and how to report is also influenced by communication difficulties between journal editors, peer reviewers and authors.[19 21] For progress to be made, better understanding of the challenges to high-quality reporting of QI is needed.

It is useful, for purposes of understanding these challenges, to bound the scope of inquiry to enable focus and depth. We selected perioperative care as an instructive area in which to examine the challenges of reporting in more depth. The volume of surgical intervention globally is huge—of the order of 313 million procedures per per annum[22] and is highly variable in quality, 4.2 million people die every year within 30 days of surgery[23] suggesting considerable room for improvement. While perioperative care has seen a huge increase in volume of literature,[24 25] it is also an area in which a systematic review that we previously conducted revealed pervasive problems of poor reporting in relation to QI.[3] With the aim of informing ways of improving QI reporting, we sought to understand the experiences, views and priorities of those involved in or influencing the publication, writing or consumption of reports of QI studies in perioperative care.

## METHODS
We used the Standards for Reporting Qualitative Research (SRQR) reporting guidelines to write this manuscript.[26]

### Study design
We undertook semistructured interviews to explore why reporting QI in perioperative care is difficult. Building on the distinction between QI interventions and methods that we had previously made,[3] we asked why reporting of QI interventions (such as checklists or care pathways) and QI methods (such as Lean or Plan-Do-Study-Act cycles) might pose challenges.

A standard set of questions was used as a basis for open discussion. The interview schedule was informed by data generated from our systematic review.[3] For example, our data showed that reporting is poor, and we included several prompts about possible explanations for this. Each interview lasted for around 45 min; most were done by telephone, with three face-to-face.

Interviews were recorded and transcribed with informed consent from participants. All data were collected between September 2015 and March 2016. This study was approved.

### Participants
We recruited an international sample of QI stakeholders working in organisations such as hospitals, universities and healthcare funding bodies. Participants were eligible for inclusion if they were willing and able to give informed consent, aged 18 years or older and had a role in QI reporting by virtue of being involved in or being an influencer of the publication, writing or consumption of reports of QI studies.

For the purposes of the study, QI authors were defined as individuals who had been an author on a paper reporting QI in perioperative care, published in a PubMed-indexed journal between 2000 and 2016. QI consumers were defined as healthcare managers and clinical staff who had read reports of QI in perioperative care and used them to inform changes in delivery of surgical care in the 24 months prior to interview. QI custodians were those who set, or sought to uphold, expectations with regard to QI reporting, and were defined as authors of reporting guidelines or journal editors who had made decisions about publishing perioperative QI papers in the 24 months prior to interview.

We used purposive non-probability sampling methods: participants were deliberately chosen with the expectation that their experience would provide relevant insights. These participants were recruited via an emailed invitation. We also advertised to recruit individuals not known to the study team using web-based publicity. Sample size was estimated based on previous studies showing that 30–40 participants was sufficient to reach theoretical saturation.[27 28]

### Data analysis
Analysis was based on the constant comparative method.[29 30] One author (EJ) initially undertook a process of open coding, supported by NVivo software, whereby she coded phrases used by interviewees in a subset of interviews that pertained to a specific idea. These codes were compared and combined into more refined thematic categories, which were then used to code the full set of interview transcripts.[30 31] A second author (GPM) read a random selection of transcripts to enhance the analysis process, by ensuring the lead author's interpretations were plausible and identifying alternative possible interpretations. This informed discussion among the authorial team to enrich the analysis and develop the insights presented in the section below.

### Patient and public involvement
Gill Penny, a patient who had experienced a complication of cardiac surgery, was engaged throughout the project to advise on the appropriateness of the interview schedule and to read a selection of transcripts, helping EJ to thematise the findings.

**Table 1** Professional groups of quality improvement (QI) authors, consumers and custodians

| | | QI author (n=15) | QI consumer (n=12) | QI custodian (n=15) | Total (n=42) |
|---|---|---|---|---|---|
| Clinical staff | Physicians: anaesthetists, internal medicine doctors, physicians, radiologists, cardiologists, surgeons | 10 | 9 | 6 | 25 |
| | Other clinicians: Nurses and allied health professionals | 1 | 2 | 0 | 3 |
| Non-clinical staff | Academic | 4 | 0 | 9 | 13 |
| | Healthcare manager | 0 | 1 | 0 | 1 |

## Findings

We invited 73 individuals to participate, of whom 42 agreed (table 1): 15 QI authors, 12 QI consumers and 15 QI custodians (11 journal editors, 4 developers of reporting guidelines). The majority of participants were from the UK (24 participants); 14 were from North America; the remainder were from Australia and mainland Europe.

Our analysis identified three major influences on reporting QI in perioperative care, and corresponding possible solutions: the broad scope of QI, challenges of reporting context and proximal and distal influences within organisations that influence QI reporting.

### The broad scope of QI as an influence on reporting

Thirty participants identified the broad scope of QI as an important influence on quality of reporting. The variety of terms used to describe QI[11 32] and the fluidity and inconsistency with which they were used were seen to interfere with clarity and precision. Some participants saw QI as defined by a strong association with specific approaches taken from manufacturing industries (eg, Lean, Six Sigma, PDSA, Statistical Process Control, and Total/Continuous Quality Management). But others felt that QI was much broader, noting the overlaps between QI and other fields, including audit, change management, human factors, implementation science, behavioural sciences, social science and engineering.

> The term 'Lean' is widely misused and used in different ways, by lots of different people, so the word doesn't necessarily have specific meanings to the reader. (QI author, anaesthetist 1)

> QI means different things to different people. (QI author, academic 1)

This plethora of terms and concepts was further complicated by ambiguity about the purpose of reports of QI. Most participants (40) distinguished between QI projects and QI research. They defined QI projects as local activities to improve the quality of care. In contrast, they defined QI research as work that uses evaluative methods, seeks generalisability or transferability, manages bias and requires ethical approval. Some participants, however, said the distinction between QI projects and QI research can be blurred, with more of a continuum than a sharp line.

> QI is more real-world and it is not research. It will inherently have all the biases. It'll have clinical biases, selection biases, reporting bias, buy-in from staff, it'll have all the biases one can think of (QI Author, Surgeon 1).

The wide range of approaches and academic disciplines involved in QI provoked uncertainty about where QI work should be published. Journal editors noted, for example, that QI authors may use a 'scattershot approach' (QI custodian, academic 1), perhaps submitting articles for publication to a wide variety of journal types.

To overcome the challenges caused by the broad scope of QI, solutions proposed by participants included: having journals dedicated to QI (11 participants); encouraging all QI stakeholders to use the SQUIRE[10] guidelines including journal editors and peer reviewers (10); having a central database of QI work in surgery (7); and a QI section in surgical journals (2).

### Challenges of reporting active ingredients and contexts in QI

Participants identified various purposes of QI reporting, which went beyond providing straightforward blueprints that could be 'dragged and dropped' to other settings. Most participants (38) recognised that not all QI work is intended to be exactly replicable, often describing features of QI that were 'transferable' from one setting to another rather necessarily 'generalisability'[33] in a broader sense. Some, especially those classically trained in experimental methods, tended to emphasise the need for generalisability, but others did not. Thus, three journal editors expected research to demonstrate generalisability to be publishable, but eight editors were happy for QI authors to explain why an intervention could be 'portable' to another setting where not all ingredients were directly reproduced.

Causal attribution was recognised by participants as a major challenge for QI. Good descriptions of interventions and methods, including their 'active ingredients', were seen as important, since most QI interventions likely

require some element of retesting in a new healthcare setting. Thus, rather than being able to 'get it off the shelf and pull it in' (QI author, academic 2), QI consumers 'use the QI publication to know what was going on and be able to adapt it (the intervention) for other settings' (QI custodian, academic 2). However, reporting the intervention was rarely seen as enough: an account of context was also required. Contextual features might include leadership, buy-in, culture, teamwork, resources and environment and many aspects of organisation and structure.[34] Participants (22) said that when the contexts of QI studies are fully reported, a greater understanding of the scope and limits of transferability to other settings can be achieved. However, it was not clear to stakeholders how features of context should be defined, or which ones should be reported. The MUSIQ tool[34] (a framework which identifies 25 contextual factors likely to influence QI) which can aid description of context already exists, but the elements of MUSIQ were discussed by only three people (two authors and one editor).

> Part of the active ingredient might have been inadvertently the culture or the attitudes of the people in the organisation which you may or may not have somewhere else. (QI custodian, academic 3)

> If you have the detail at least you can see that it is something we could do in our location…Knowing the detail can allow for assessment. (QI author, cardiologist)

The emphasis on the importance of context was accompanied by recognition among all 42 participants of the difficulties of reporting it. Many participants (28) reported the basic problem that it is difficult to characterise what is meant by context and to distinguish it precisely from intervention. Sometimes context was described as amorphous and ethereal, akin to a black box. For example, corridor conversations, chance meetings of charismatic personalities, a changed team member, simultaneous work in other departments or board-level decisions can critically affect the outcome of a QI project for better or worse, but these occurrences may evade capture.

Many participants (17) were ambivalent about drawing a hard line between context and intervention given that both might be implicated in change, yet how to describe this was not clear. Some noted that some ingredients may be more active in one place than another, and it can be hard to work out 'which are the most important ingredients with the greatest weight' (QI author, surgeon 1). Thus, not only was it difficult to identify contextual features, it was reported that it is also hard to determine which ones are important (10 participants).

Despite the emphasis given to context, some participants reported that contextual features were at risk of being seen, particularly by more epidemiologically trained editors and reviewers, as 'noise' that should be 'controlled out'. These participants characterised contextual features as confounders, sources of bias (which

systematically influence the direction a QI study takes) or natural variation (factors that are happening anyway, over which the researcher has no control). Four journal editors were concerned that when authors seek to explain contextual features that are specific to individual localities, peer reviewers might then suggest that further evaluation in new settings was needed, making it harder to publish QI work.

> The reality is that how this project will play out in a different hospital is different because of a whole bunch of idiosyncratic workflow issues. And so even if it worked in this one hospital, it's almost like, anyone else wanting to do it is going to have to redo it. There's so many different ways in which even something as basic as a checklist can be done, they're going to essentially have to do the same thing the authors did (QI custodian, doctor).

A particular challenge in reporting context was that some members of the scientific community may fail to value qualitative methods, even though they may be especially well suited to describing context. Participants reported that some authors might 'roll their eyes' (QI author, surgeon 2) when asked to report context because they do not have the skill to report it, cannot specify it or cannot fit it into conventional models of reportage: it feels like 'fitting a square peg in a round hole' (QI author, surgeon 3). Further, negative contextual features (such as bullying or seeking to sabotage interventions) were seen as difficult to describe candidly (13 participants).

> All under the carpet…people don't want to say the chief of surgery was an idiot and we had to get the hospital president to sit, make him agree to this [QI research]. (QI custodian, surgeon)

To improve understanding of how to report context, participants suggested: wider use of the MUSIQ tool[34] (12 participants); extending the MUSIQ tool to highlight contextual features known to affect QI in surgery (3); including the study of context in medical school curricula on QI (6); and using terms such as 'portable' and 'reproduce' in lieu of 'generalisable' and 'replicate' to encourage understanding that not all mechanisms contributing to an intervention's success or failure can be replicated exactly, and some interventions (and contextual features) may need to be adapted to other settings (14). Participants suggested that collection of contextual data may be eased by use of: objective scales (8); QI diaries kept by the researchers, which participants likened to lab books (6); external independent evaluation (3); and ethnography (2). As some participants found it hard to report context in the conventional journal format, eight participants suggested adding a heading of 'What really happened'. Eight, however, did not want the traditional introduction, methods, results and discussion (IMRAD) structure of academic papers to be altered to better suit QI reporting.

The traditional journal format is established, it has a tremendous amount of weight and is respected and successful, and I think if quality improvement can sit in that model it should. (QI author, radiologist)

### Proximal and distal influences leading to incomplete reporting of QI

Participants described how certain features of the organisational and institutional fields in which they worked might influence the quality of reporting. Many participants (10 QI authors, 5 QI consumers and 8 QI custodians, all in mixed clinical and academic roles) noted how personal or organisational self-interest might prompt QI authors to seek to publish their work, given its potential implications for allocation of research funding[35][36] and in performance management and reputation.[35]

Personal credit, ambition, glory… (QI author, anaesthetist 2)

Thirty-four participants referred to the potential benefit for patients as a principal motivation for reporting QI work, and 17 sought to reduce wasteful duplication across healthcare sites. But many participants also reported barriers and disincentives to reporting. We conceptualise these barriers as proximal (close to the writing-up activities of QI authors) and distal (related to higher-level organisational and institutional influences).

A third of participants (14) discussed proximal barriers—for example, the challenge of doing QI work and writing it up, while simultaneously looking after patients. This might be particularly challenging for authors who conduct QI alongside everyday patient care, perhaps in contrast to those on clinical academic pathways who may have allocated time for QI research. Participants used phrases like 'feeling battered' (QI consumer, anaesthetist 1), 'on a hamster wheel' and 'wading through treacle' (QI consumer, anaesthetist 2). Even if they could find time to produce complete QI reports, six participants felt that what they report (selecting the QI topic and deciding which contextual features should be included) was influenced by their immediate hospital management. Twenty-two participants reported mundane, practical challenges—for example, how writing can be hampered by restrictive word counts. Similarly, some suggested that reporting guidelines might have only a limited role in improving the quality of reporting, especially if QI stakeholders do not realise they exist (10 participants).

One of the reviewers said we hadn't used any guidelines, even though we'd used SQUIRE, but he had never come across it before. (QI author, physiotherapist)

These proximal barriers were often profoundly structured by distal influences—for example, the norms surrounding article format that are widely accepted within the field of biomedical research, and which also inform expectations for publishing studies of QI. Publishing in high-impact journals was seen as challenging of a perceived preference for quantitative data over qualitative explanations of contextual features (8), and/or focus on novel therapeutic approaches (13)—neither of which favour QI. Nineteen participants also reported that explaining failure may be so difficult that negative or null QI studies may never be written up or published. When asked what authors find most difficult to write about in QI, one participant responded:

Stuff that didn't work! [laughs]…I think publishing null studies is always hard and a lot of people don't do it. (QI author, surgeon 4)

Participants proposed several solutions to these challenges. Heavy clinical workloads that perpetuate poor reporting could be alleviated by: allowing protected time for QI work (3); convening multidisciplinary writing teams (14); embedding local or regional QI research units that could operate in the same way as clinical trials units (6); providing structured programmes of QI education or mentorship (13); and involving patients, who could also be part of a QI multidisciplinary team (7).

If you're thinking of sort of blue sky, I can imagine that you know, in the very same way as we have clinical trials [units] we should have quality improvement units. (QI custodian, academic 3)

Participants generally felt that word counts should not be increased, because brevity is valued in scientific writing, but the constraints they impose could be alleviated by: uploading supplementary material and podcasts (18); encouraging multiple publications for a single QI study (4); using web-enabled formats that allow the reader to explore topics in more depth depending on what they are most interested in (15); and sections dedicated to negative studies in journals (2). Any solutions proposed would need, however, to be implemented through agreement among journal editors (13).

The editors should be the ones who need to really drive this to make sure enough detail is included in the papers. (QI author, cardiologist)

## DISCUSSION

This study of stakeholders' views on what influences QI reporting in perioperative care suggests that its fit with traditional forms of scientific research is imperfect, and the rules and norms that govern QI authors' and QI custodians' understanding of what is worth publishing are not always aligned.

Some reasons why reporting QI is so hard are potentially tractable, but will require both maturing of the field and convergence between the views of different stakeholders.[37] One challenge for QI reporting is that the contextual features which are important in mitigating failure or facilitating success are critical to the fabric of the QI work,[38–42] but we found little consensus on how best to report context. In a field of study that remains young, this is perhaps not surprising, and there is a need for further

**Table 2** Improving the reporting of QI in surgery: approaches suggested by interview participants

| Domain | Potential actions for QI authors | Potential actions for healthcare organisations delivering QI work | Potential actions for journal editors publishing QI work |
|---|---|---|---|
| Article format | Use existing reporting guidelines and taxonomies to guide the structure of your QI report. | | Ensure familiarity of editorial staff and peer reviewers with QI reporting tools. |
| | Know your audience. Do you want your reader to use the report to generate ideas for a new intervention, to replicate your intervention in another setting, or as a starting point for modification? | | Provide a clear statement about whether qualitative approaches to data collection and writing are acceptable. |
| | Use supplementary materials, and embed URLs (web links) into the article where possible. | | Provide a clear statement of which additional resources are available to authors (eg, online supplements). |
| | Be available to speak to your readers | | Support the open access movement to encourage connection between authors and consumers. |
| Organisational infrastructure | | Build internal support and capacity for QI, such as protected time to conduct QI and more formal relationships between clinical QI teams and research nurses. | Sustain open communication channels with QI authors and consumers about what QI is and how it should be reported. |
| | Consider using a multidisciplinary writing team, how to support patient involvement, and seeking external evaluation. | Build networks with external academic organisations (such as universities) and patients. | |
| | Work with hospital management to identify problems that are most relevant to patients (enable a breadth of topics). | Work with QI teams to identify problems that are most relevant to patients (enable a breadth of topics). | |
| | Consider enrolling in an education programme to enhance your QI reporting. | Embed specific training about QI in library training programmes, online training programmes or mentorship schemes. | Consider providing some educational material for editors and peer reviewers about QI. |
| Scientific outputs | Demonstrate why your intervention was thought to work (eg, consider using theory, process evaluation, or a QI diary). | | Enable structured conversations with QI stakeholders to consider how QI can be reported and what good reporting in QI looks like. |
| | Provide your reader with a realistic view of what is needed and what is feasible. | | |
| | Consider submitting for publication a QI project that did not go well. | Support a culture where negative experiences that create learning are shared. | Give specific advice on how to write a negative study well. |

*Taxonomy and Reporting guideline examples.[10 34 51–53]
QI, quality improvement.

consideration of this issue within the QI community. Our findings indicate that the range of QI interventions and contexts, as well as the diversity of reasons for publishing QI and associated intended audiences, means that what should and should not be reported is not readily reducible to universal criteria.

The proximal pressures we have outlined (such as QI authors not having enough time to treat patients and write up QI work, and the mismatch between the norms of biomedical publication and the expectations of QI authors) could potentially be relieved by practical support. Emergent models of research and practice in improvement may help to bring researchers and practitioners together, carving out time for reporting and ensuring the relevance of research for QI practitioners. Recent developments such as researcher-in-residence

models,[43] boundary-spanning roles[44] and the like have some promise in generating important insights for busy clinicians that may otherwise remain uncovered, while also yielding publications of greater relevance and usefulness.[45]

The more distal influences on reporting that we have identified point towards the issues that underlie some of the symptoms of the challenges of QI reporting, and which may be harder to shift. The forms of knowledge that are valued within clinical–academic circles, for example, are underpinned by enduring assumptions about the validity of knowledge, as well as incentive systems that view some forms of research and some forms of reporting as more worthwhile than others. Similarly, making more time available for QI activity and QI research may require significant shifts in the priorities of the funders of both healthcare provision and healthcare research.

Nevertheless, participants offered a range of suggestions about how these issues might be addressed, which we summarise in table 2. Some may already be in use—for example, many authors already use the SQUIRE guidelines.[10] Others are more aspirational, and may be challenging to enact. For example, QI authors in surgery may experience conflicting demands on their time, and face competing demands for brevity in scientific writing and for a full description of a social process. As such, some of them are less solutions, and more areas where coordinated attention might benefit the field. Attempts to retain what is valued about QI while continuing to satisfy a deeply ingrained way of working will require concerted and perhaps entrepreneurial efforts.[46]

The field of perioperative care was chosen as a focus of this study in part because of the evidence that it is a highly active site of QI, but demonstrates poor quality reporting.[3] It is increasingly clear that high quality science will be needed to support improvement,[47] and better reporting will be an essential element of this. Given the continued interest in developing the field of perioperative care, it is possible that targeted efforts to support improvement in reporting could yield significant benefits.

This is among the first studies to examine the challenges of reporting QI in the perioperative literature and how QI reporting might be improved. We opted not to include all types of stakeholders in QI reporting—in particular, patients. This is because public and patient involvement in QI reporting is early in its development.[48 49] Researchers may need to improve the reporting standards of QI itself[3] and allow time for the SQUIRE guidelines to become optimally implemented[50] before attempts are made to add public and patient involvement to reporting requirements. A further limitation is that we recruited only stakeholders who were actively interested in QI. However, a semistructured interview schedule and many open-ended probes allowed us to obtain a range of views.

## CONCLUSION

The fit between QI reporting and reporting of more traditional medical research poses problems for those seeking to report on QI activity, and QI custodians need to work with QI authors and QI consumers to develop more appropriate approaches. Participants had numerous suggestions about how to address such challenges (table 2), but many of these will require coordinated effort within the QI community, and should be taken forward with caution given their potential downsides and unintended consequences. Perioperative care may be a useful area in which to test some approaches.

**Acknowledgements** The authorship team would like to thank Katrina Brown, Communications Manager, THIS Institute for her help in editing and proof reading this document and Gill Penny, our patient advisor.

**Contributors** EJ initiated the idea for the interview study and led the development of the protocol, study administration, data collection, analysis and writing of this manuscript. GPM and MDW contributed to protocol development, supervised the study and contributed to data analysis and writing of the manuscript.

**Funding** This work was completed as part of a PhD studentship funded by the Health Foundation. Writing up of this paper was supported in part by MDW's Wellcome Trust Senior Investigator award WT097899. GPM acknowledges the support of the National Institute for Health Research (NIHR) Collaboration for Leadership in Applied Health Research and Care East Midlands (CLAHRC EM). MDW and GPM are supported by the Health Foundation's grant to the University of Cambridge for The Healthcare Improvement Studies (THIS) Institute. THIS Institute is supported by the Health Foundation–an independent charity committed to bringing about better health and health care for people in the UK. MDW is a National Institute for Health Research (NIHR) Senior Investigator (NF-SI-0617-10026). The views expressed in this article are those of the authors and not necessarily those of the NHS, the NIHR, or the Department of Health and Social Care.

**Disclaimer** The views expressed in this article are those of the authors and not necessarily those of the NHS, the NIHR or the Department of Health and Social Care.

**Competing interests** None declared.

**Patient consent for publication** Not required.

**Provenance and peer review** Not commissioned; externally peer reviewed.

**Data sharing statement** All data relevant to the study are included in the article - All authors had access to all the data in the study and take responsibility for the integrity of the data and the accuracy of the data analysis. No additional data are available. No unpublished data are available outside of the study team.

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
