## [Reviewer comments · BMJ Open]

ARTICLE DETAILS

TITLE (PROVISIONAL)	Why is reporting quality improvement so hard? A qualitative study in perioperative care
AUTHORS	Jones, Emma; Dixon-Woods, Mary; Martin, Graham

VERSION 1 – REVIEW

REVIEWER	Dr. Carl Savage, PhD Karolinska Institutet Medical Management Centre Dept. of Learning, Informatics, Management & Ethics Sweden
REVIEW RETURNED	27-Mar-2019

GENERAL COMMENTS	Methods Fitting choice of reporting framework. Not quite sure about the differentiation between method and intervention. One could argue that lean is a strategy and a philosophy that utilizes a number of methods, e.g. A3, and that an effort to implement a lean strategy could be seen as an intervention. I think the key question that the reader wonders is why this differentiation is important and what potential impact does this differentiation have on the data collection, analysis, and conclusions? Did the literature review provide any guidance on the methods? For example, selection of participants. This is in reference to the statement that opens the last paragraph of the introduction, "Building on the findings of an earlier systematic review". Data analysis: is it that codes were added to the phrases or that the phrases were coded? Patient and public involvement: I think it wonderful to read that a patient was involved, but I do wonder why, if the purpose was to ensure relevance for those interested in QI reporting, which suggests a "consumer" role, patients weren't included in that participant group? Why patients weren't included in the study is explained at the end of the discussion. Perhaps the last sentence of the paragraph should therefore be struck. In terms of analysis method, I am unclear as how to interpret this paragraph – does the procedure described suggest the potential that an individual with their own personal interests and biases could sway the analysis based on what they found to be interesting? In some ways this strikes me as the antithesis of why
---

	we use reflexivity in qualitative analysis. Perhaps a description of the reflections that arose from discussing the themes might be appropriate as an additional data source/analysis? Would it make sense to include a couple of brief sentences on reflexivity related to the patient/member of public as well as the authors? Findings A reflection that might be of interest to comment on in the discussion: Was there any connection between the type of analysis/data collected and used by the participants (qualitative vs quantitative) and their preference on “transferability” vs. “generalizability”? I.e. is the choice of word a question of (e.g. academic) schooling or of function (e.g. consumer) – that they are looking to transfer what they learned whereas a researcher might be more interested in generalizing and stopping there. I assume the numbers in parentheses refer to the number of participants who were coded in that category? Not sure this adds much since the analysis is not explicitly described as using a summative approach nor are the numbers discussed in the discussion. Introduction (some minor comments) Introduction, first paragraph: Not sure about the examples used to exemplify the differentiation between methods and interventions. For example, cannot lean be an intervention? Whereas an A3 might be a method used in a lean intervention. By the same token, a checklist can be an intervention, e.g. implementation of the WHO Safe Surgery Checklist, or a method used in, for example, a CRM intervention to improve team communication and patient safety. Introduction, third paragraph: Would be good to include the “S” part of the SQUIRE abbreviation in the sentence. Introduction, third paragraph: Might also be interesting to include a comment on why SQUIRE went from 1.0 to 2.0, which could further function as a description of the practical problem, but I leave that to the discretion of the authors. Introduction, last paragraph: While it may make sense to limit the scope of the area under review, i.e. perioperative care, It is unclear why the authors chose this area beyond the fact the en passant reference to the literature review. A more clear motivation would strengthen the study rationale. Tables Ok. Perhaps include a list of potential guidelines as recommendations to authors?
--	---

REVIEWER	Elizabeth Wick UCSF; USA
REVIEW RETURNED	24-Apr-2019

GENERAL COMMENTS	This is well done study that is important. The authors selected to focus on the perioperative area. This seems to be glanced over -- it would be great to elaborate more in the discussion any findings that seemed unique to periop as compared to other areas. Of the folks interviewed, some of the quotes are attributed to surgeons but in table 1, there is no note of a surgeon in the clinical staff. Were there any priorities of context (leadership; culture; accountability; data etc) that came through as things that should be consistently reported? One of the challenges of context is the selective reporting.
--

VERSION 1 – AUTHOR RESPONSE

Reviewer 1 comments:

REVIEWER COMMENTS	AUTHORS' RESPONSE	Where located
TABLES: Ok. Perhaps include a list of potential guidelines as recommendations to authors?	We have added the following as a footnote (all added text shown in red) to table 2. We feel it is best not to include this as a separate table, since our findings suggest that reporting guidelines will be at most only part of the solution. *Taxonomy and Reporting guideline examples:  • Hoffmann TC, Glasziou PP, Barbour V, Macdonald H. Better reporting of interventions: template for intervention description and replication (TIDieR) checklist and guide. BMJ. 2014;348:g1687(March):1–13. • Kaplan HC, Provost LP, Froehle CM, Margolis PA. The Model for Understanding Success in Quality (MUSIQ): building a theory of context in healthcare quality improvement. BMJ Qual Saf. 2012 Jan;21(1):13–20. • Ogrinc G, Davies L, Goodman D, Batalden P, Davidoff F, Stevens D. SQUIRE 2.0 (Standards for QUality Improvement Reporting Excellence) : revised publication guidelines from a detailed consensus process. BMJ Qual Saf. 2016 Dec 25 (2) 986 – 992. • Pinnock H, Barwick M, Carpenter CR, Eldridge S, Grandes G, Griffiths CJ et al Standards for Reporting Implementation Studies(StaRI) Statement. BMJ. 2017 Mar 6. 	See table 2

	 Schulz, R; Czaja, S; McKay, J; Ory, M; Belle S. NIH Public Access Intervention Taxonomy (ITAX): Describing Essential Features of Interventions (HMC). Am J Heal Behav. 2010;34(6):811–21. 	
INTRODUCTION: While it may make sense to limit the scope of the area under review, i.e. perioperative care, It is unclear why the authors chose this area beyond the fact the en passant reference to the literature review. A more clear motivation would strengthen the study rationale.	Thank you. We have amended the text as follows to explain why we focused on perioperative care (all text added is shown in red): It is useful, for purposes of understanding these challenges, to bound the scope of inquiry to enable focus and depth. We selected perioperative care is an instructive area in which to examine the challenges of reporting in more depth. The volume of surgical intervention globally is huge – of the order of 313 million procedures per per annum²² -- and is highly variable in quality: 4.2 million people die every year within 30 days of surgery²³suggesting considerable room for improvement. While perioerative care has seen a huge increase in volume of literature,^{24,25} it is also an area in which a systematic review³ that we previously conducted revealed pervasive problems of poor reporting in relation to QI. ³With the aim of informing ways of improving QI reporting, we sought to understand the experiences, views and priorities of those involved in or influencing the publication, writing, or consumption of reports of QI studies in perioperative care.	Introduction , last para
Might also be interesting to include a comment on why SQUIRE went from 1.0 to 2.0, which could further function as a description of the practical problem, but I leave that to the discretion of the authors.	Thank you, we agree that this is an interesting point but we are reluctant to expand on this in order to give priority to other comments which have been added, and ensure the word count does not increase excessively.	No change to manuscript
Would be good to include the “S” part of the SQUIRE abbreviation in the sentence.	Corrected	Line 21
Not sure about the examples used to exemplify the differentiation between methods and interventions. For example, cannot lean be an intervention? Whereas an A3 might be a method used in a lean intervention. By	Again, thank you very much for your careful consideration of our paper and raising an important point. We do not feel we can do justice to this distinction within this manuscript. However, one of the authors (EJ) has coauthored another manuscript which addresses this issue. The manuscript has been accepted (subject to minor revisions) by BMJ Open Quality. For the purposes of this manuscript, we have altered the paragraph in question to add the caveat ‘for the	Line 4

the same token, a checklist can be an intervention, e.g. implementation of the WHO Safe Surgery Checklist, or a method used in, for example, a CRM intervention to improve team communication and patient safety	purposes of our systematic review we previously defined QI as involving...'. The paragraph now reads (with text added in red): We previously, in the context of a systematic review³, defined QI as involving both QImethods, including approaches such as Plan-Do-Study-Act cycles, Lean, and Six Sigma,^{1,4} and QI interventions, which are specific activities, actions or instruments targeting defined areas of practice (e.g. checklists).⁵	
METHODS: Not quite sure about the differentiation between method and intervention. One could argue that lean is a strategy and a philosophy that utilizes a number of methods, e.g. A3, and that an effort to implement a lean strategy could be seen as an intervention. I think the key question that the reader wonders is why this differentiation is important and what potential impact does this differentiation have on the data collection, analysis, and conclusions?	As above - In a second paper which has just been accepted (pending revisions) by BMJ Open Quality written by Emma Jones with Joy Furnival and Wendy Carter, we consider length this potentially challenging distinction between interventions and methods and the problems with checklists and lean. We have now amended the text to add the text in red as follows. Study design We undertook semi-structured interviews to explore why reporting QI in perioperative care is difficult. Building on the distinction between QI interventions and methods that we had previously made,³ we asked why reporting of QI interventions (such as checklists or care pathways) and QI methods (such as Lean or Plan-Do-Study-Act cycles) might pose challenges.	Line 54
Did the literature review provide any guidance on the methods? For example, selection of participants. This is in reference to the statement that opens the last paragraph of the introduction, "Building on the findings of an earlier systematic review".	Yes, the literature review was helpful in guiding the structure of our interview schedule and we have sought to highlight this more clearly by adding the text in red to this paragraph: A standard set of questions was used as a basis for open discussion. The interview schedule was informed by data generated from our systematic review [REF]. For example, our data showed that reporting is poor, and we included several prompts about possible explanations for this. Each interview lasted for around 45 minutes; most were done by telephone, with three face-to-face.	Line 58
Data analysis: is it that codes were added to the phrases or that the phrases were coded?	The phrases were coded. We have now explained this.	Line 90
Patient and public involvement: I think it wonderful to read that a patient was involved, but I do wonder why, if the	We agree with this point and we have deleted the last sentence of the PPI paragraph. Thank you.	Line 102

purpose was to ensure relevance for those interested in QI reporting, which suggests a “consumer” role, patients weren’t included in that participant group? Why patients weren’t included in the study is explained at the end of the discussion. Perhaps the last sentence of the paragraph should therefore be struck.		
In terms of analysis method, I am unclear as how to interpret this paragraph – does the procedure described suggest the potential that an individual with their own personal interests and biases could sway the analysis based on what they found to be interesting? In some ways this strikes me as the antithesis of why we use reflexivity in qualitative analysis. Perhaps a description of the reflections that arose from discussing the themes might be appropriate as an additional data source/analysis? Would it make sense to include a couple of brief sentences on reflexivity related to the patient/member of public as well as the authors?	Our intention in adopting this approach was not to try to achieve objectivity in our analysis, but to ensure that the lead author’s interpretations and inferences from the data were plausible and defensible. Accordingly, we have rephrased the final sentence of this section, and include a fuller account of how we used it to inform the construction of our findings (text added is shown in red): “A second author (GM) read a random selection of transcripts to enhance the analysis process, by ensuring the lead author’s interpretations were plausible, and identifying alternative possible interpretations. This informed discussion among the authorial team to enrich the analysis and develop the insights presented in the section below.”	Line 94
FINDINGS/DISCUSSION A reflection that might be of interest to comment on in the discussion: Was there any connection between the type of analysis/data collected and used by the participants (qualitative vs quantitative) and their	This is an interesting point. We have added another few sentences which reflect the data presented to explain why some participants leaned towards generalisability, rather than ‘transferability’: ADDED TEXT SHOWN IN RED: Participants identified various purposes of QI reporting, which went beyond providing straightforward blueprints that could be ‘dragged and dropped’ to other settings. Most participants (38)	Line 152

preference on “transferability” vs. “generalizability”? I.e. is the choice of word a question of (e.g. academic) schooling or of function (e.g. consumer) – that they are looking to transfer what they learned whereas a researcher might be more interested in generalizing and stopping there.	recognised that not all QI work is intended to be exactly replicable, often describing features of QI that were “transferable” from one setting to another rather necessarily “generalisability”³³ in a broader sense. Some, especially those classically trained in experimental methods, tended to emphasise the need for generalisability, but others did not. Thus, three journal editors expected research to demonstrate generalisability to be publishable, but nine editors were happy for QI authors to explain why an intervention could be ‘portable’ to another setting where not all ingredients were directly reproduced.	
I assume the numbers in parentheses refer to the number of participants who were coded in that category? Not sure this adds much since the analysis is not explicitly described as using a summative approach nor are the numbers discussed in the discussion.	Although the numbers are not explicitly discussed, we have left them in for transparency so that the reader can judge their importance on the weight of the data for themselves. This is consistent with the advice of Seale (Seale, C. F. (1999). The quality of qualitative research. London: Sage.) Therefore, the manuscript has not been amended.	No changes to manuscript

Reviewer 2 comments:

REVIEWER 2	AUTHORS' RESPONSE	Where located
The authors selected to focus on the perioperative area. This seems to be glanced over -- it would be great to elaborate more in the discussion any findings that seemed unique to periop as compared to other areas.	Thank you. The same point was raised by Reviewer 1 and we have addressed in in the Introduction. We are limited in how far we can discuss any findings that are unique to periop as we have not conducted research in other fields for the present. However, we have added the following text to the Discussion: The field of perioperative care was chosen as a focus of this study in part because of the evidence that it is a highly active site of QI, but demonstrates poor quality reporting.³ It is increasingly clear that high quality science will be needed to support improvement, and better reporting will be an essential element of this. Given the continued interest in developing the field of perioperative care, it is possible that targeted efforts to support improvement in reporting could yield significant benefits.	Line 352
Of the folks interviewed, some of the quotes are attributed to surgeons but in table 1, there is no note of a surgeon in the clinical staff.	Many thanks for your careful consideration of our paper and apologies for this oversight. While considering how best to present this data to ensure anonymity, and to ensure it could be read by a wide international audience we used the term physicians (to cover surgical and medical practitioners), but we realise this might be read as	Table 1

	conflating two distinct groups. We have sought to clarify this point in table 1.	
Were there any priorities of context (leadership; culture; accountability; data etc) that came through as things that should be consistently reported? One of the challenges of context is the selective reporting.	The participants discussed context at length but they found it difficult to explain or elaborate on particular features of context. We have added text to the existing paragraph below in red, which reflects the data presented: Causal attribution was recognised by participants as a major challenge for QI. Good descriptions of interventions and methods, including their “active ingredients”, were seen as important, since most QI interventions are likely require some element of re-testing in a new healthcare setting. Thus, rather than being able to “get it off the shelf and pull it in” (QI author, academic 2), QI consumers “use the QI publication to know what was going on and be able to adapt it [the intervention] for other settings” (QI custodian, academic 2). However, reporting the intervention was rarely seen as enough: an account of context was also required. Contextual features might include leadership, buy-in, culture, teamwork, resources, and environment and many aspects of organisation and structure.³⁰ Participants (22) said that when the contexts of QI studies are fully reported, a greater understanding of the scope and limits of transferability to other settings can be achieved. However, it was not clear to stakeholders how features of context should be defined, or which ones should be reported. The MUSIQ tool (a framework which identifies 25 contextual factors likely to influence QI) which can aid description of context already exists, but the elements of MUSIQ were discussed by only three people (two authors and one editor).	Line 173